# UPLC-MS Analysis, Quantification of Compounds, and Comparison of Bioactivity of Methanol Extract and Its Fractions from Qiai (*Artemisia argyi* Lévl. et Van.)

**DOI:** 10.3390/molecules28052022

**Published:** 2023-02-21

**Authors:** Ting Zhang, Dingrong Wan, Yuanyuan Li, Sisi Wang, Xiuteng Zhou, Fatemeh Sefidkon, Xinzhou Yang

**Affiliations:** 1School of Pharmaceutical Sciences, South-Central Minzu University, Wuhan 430074, China; 2State Key Laboratory of Dao-di Herbs, National Resource Center for Chinese Materia Medica, China Academy of Chinese Medical Sciences, Beijing 100700, China; 3Research Division of Medicinal Plants, Research Institute of Forests and Rangelands, Agricultural Research Education and Extension, Organization (AREEO), Tehran 13185-116, Iran

**Keywords:** *Artemisia argyi* Lévl. et Van., UPLC-MS, quantification, anti-inflammatory, antioxidant

## Abstract

The *Artemisia argyi* Lévl. et Van. growing in the surrounding areas of Qichun County in China are called Qiai (QA). Qiai is a crop that can be used both as food and in traditional folk medicine. However, detailed qualitative and quantitative analyses of its compounds remain scarce. The process of identifying chemical structures in complex natural products can be streamlined by combining UPLC-Q-TOF/MS data with the UNIFI information management platform and its embedded Traditional Medicine Library. For the first time, 68 compounds in QA were reported by the method in this study. The method of simultaneous quantification of 14 active components in QA using UPLC-TQ-MS/MS was reported for the first time. Following a screening of the activity of QA 70% methanol total extract and its three fractions (petroleum ether, ethyl acetate, and water), it was discovered that the ethyl acetate fraction enriched with flavonoids such as eupatilin and jaceosidin had the strongest anti-inflammatory activity, while the water fraction enriched with chlorogenic acid derivatives such as 3,5-di-O-caffeoylquinic acid had the strongest antioxidant and antibacterial activity. The results provided the theoretical basis for the use of QA in the food and pharmaceutical industries.

## 1. Introduction

*Artemisia argyi Lévl.* et Van. is widely distributed in East Asian countries, especially in China. *Artemisia argyi* is a common flavoring and colorant in the food industry, and also a traditional medicine used to manage dysmenorrhea and inflammation [1]. Another use is in moxibustion, a form of traditional Chinese medicine that involves burning the plant materials over acupuncture points [2]. The mugwort grown in Qichun County, Hubei Province, China, is called “Qiai”. According to Li Shizhen’s “*Compendium of Materia Medica*”, a classical Chinese medicine work, the quality of Qiai is superior to other regions [3]. Modern studies suggest that Qiai contains a wide range of active ingredients, including phenolic acids, terpenes, polysaccharides, and essential oils [4,5,6]. Furthermore, the essential oil, tannins, and flavonoid concentration in Qiai are higher than in other production areas [7,8,9]. Although the prices of Qiai are higher than in other production areas, its demand remains robust. As research progresses, the pharmacological effects of *Artemisia argyi*, such as anti-inflammatory [10], anti-tumor [11], and obesity improvement [12], become clearer, and more and more *Artemisia argyi* products are developed and utilized [13]. By 2021, the planting area in Qiai reached 20,000 hectares, with an industrial output value of 1.16 billion dollars.

Phenolic compounds, as the main components in QA, have successfully attracted the attention of most researchers [14,15]. Most studies have made attempts in recent years to indicate the bioactivity of QA’s total phenolic compounds. However, nothing is currently known regarding the qualitative and quantitative analyses of the phenolic compounds in QA. Only 18 phenolic acids were preliminarily identified by ultra-high-performance liquid chromatography quadrupole time-of-flight mass spectrometry (UPLC-Q-TOF/MS) [16], 10 phenolic acids were identified and 7 phenolic acids quantified by HPLC [17], and 6 volatile compounds were detected by GC-MS [18]. The complexity of phytochemistry influenced the quantitative results in their research. Their results are deemed inadequate for the ongoing study of QA, so it is necessary to analyze and detect the phenolic components accurately and systematically. An attempt has been made to combine UPLC-Q-Exactive-MS/MS with mass spectrometry databases such as MZVault, MZCloud, and BGI Library for the preliminary identification of 125 chemical components in mugwort leaves from Henan Province, showing that combining UPLC-MS with a phytoconstituent mass spectrometry database can greatly improve the efficiency of compound characterization [19]. In this study, the combination of UPLC-Q-TOF/MS with the UNIFI platform enables rapid and automatic characterization of chemical constituents in plants, which has the advantages of high sensitivity, good selectivity, and easy operation [20]. An efficient qualitative method allowed us to identify 68 phenolic compounds from 70% methanol total extract of QA (QA-TE) by combining UPLC-Q-TOF/MS and the UNIFI platform.

Previous studies have shown that phenolic compounds have various pharmacological activities [21,22]. However, it is not clear which chemical components are responsible for these pharmacological activities. Bioassay data showed that the QA-TE and its water fraction (QA-FWT) had good antioxidant activities, and the ethyl acetate fraction (QA-FEA) and water fraction (QA-FWT) had favorable anti-inflammatory and antibacterial activities. Through further accurate quantitative analysis of the total extract and fractions by UPLC-TQ-MS/MS with superior sensitivity and stability [23], it was revealed for the first time that the antioxidant activity of QA was attributed to phenolic compounds, the anti-inflammatory activity was attributed to flavonoids, and the antibacterial activity was attributed to chlorogenic acid derivatives.

To the best of our knowledge, this is the first study using UPLC-Q-TOF/MS combined with the UNIFI data platform to quickly characterize compounds in QA, and the first study using UPLC-TQ-MS/MS to quantify compounds in QA. Therefore, this work will contribute to the availability of more references for the characterization and quantification of compounds in QA. Beyond that, the work will facilitate providing a theoretical foundation for the application of QA in food, pharmaceutical, and other industries.

## 2. Results and Discussion

### 2.1. Identification of QA Extract by UPLC-Q-TOF/MS

The QA extract solution was detected using UPLC-Q-TOF/MS technology under chromatographic and mass spectrometry conditions. The rapid, efficient and validated UPLC-Q-TOF/MS analytical method was established for the identification of the main chemical components in QA. The base peak ion chromatograms (Figure 1) provide the metabolomic analysis, also known as the analytical fingerprint for plant identification and authentication, a fairly integrated frame.

The collected MS data were imported into the UNIFI information management platform. In the UNIFI information software, the theoretical database of QA leaf compounds and the physical database of reference substances were established. As shown in Table 1 and Table 2, a total of 68 compounds were identified in QA leaves, with 47 compounds identified by positive ion mode collection and 43 compounds identified by negative ion mode (22 compounds were collected by both positive and negative ions). This is the first time that the combination of UPLC-Q-TOF/MS and the UNIFI platform has been applied to characterize the compounds in QA, and the established method has successfully identified the largest number of compounds. Among these are well-known phytochemicals, such as chlorogenic acid, jaceosidin, eupatilin, quercetin, and 3,5-di-O-caffeoylquinic acid, which possess antioxidant, anti-inflammatory, cancer chemopreventive, immunosuppression, and food additive properties.

### 2.2. Quantitative Analyses of Fourteen Compounds

#### 2.2.1. Method Validation

Figure 2 depicts the representative UPLC-TQ-MS/MS total ion chromatogram of standards, QA-TE, QA-FEA, and QA-FWT. Figure 3 depicts the ion chromatograms of 14 standards under the optimal UPLC-TQ-MS/MS conditions. The method’s linearity, sensitivity, precision, and accuracy satisfy international standards. The linearity of the standard solution was assessed by analyzing the standard solution over a concentration range satisfactory for the quantification of the relevant analytes in the sample. All analytes’ regression equations had excellent linearities, with the determination coefficient R^2^ ≥ 0.9967 (Table 3). All analyte detection limits ranged from 0.48 to 5.32 ng/mL (Table 3), while all analyte quantitation limits ranged from 1.45 to 15.89 ng/mL (Table 3). To the best of our knowledge, this is the lowest limit of the quantification method for the simultaneous quantification of compounds in QA. Additionally, for the peak region of all analytes, the intra-day and inter-day RSDs were less than 2.31% and 2.16%, respectively (Table 3).

These findings demonstrate that the approach has good precision whether used to measure on an intra-day or day-to-day basis. Additionally, the range of spiking recoveries for all analytes was 99.79% to 104.37% (Table 3), demonstrating that the method has adequate accuracy. Furthermore, the analyte recovery range was measured to be 97.56% to 101.74% (Table 3). The findings indicate that the adopted methodology has good linearity, sensitivity, precision, accuracy, and stability, and can be used to quantify fourteen characteristic compounds from QA leaves.

#### 2.2.2. Quantitative Analysis

The developed UPLC-TQ-MS/MS method was subsequently applied to quantify 14 bioactive compounds in leaves of A. argyi. Table 4 shows the quantification results for extracts and fractions. The *p*-values for all compounds measured were less than 0.05. Figure 4 depicts the structures of quantified compounds in Qiai. The quantified compounds belonged to two classes, eight flavonoids (chrysoeriol 7-O-glucoside, chrysoeriol, schaftoside, isoschaftoside, hyperoside, hispidulin, eupatilin, and jaceosidin) and six chlorogenic acid derivatives (3,5-di-O-caffeoylquinic acid, 3,4-di-O-caffeoylquinic acid, 4,5-di-O-caffeoylquinic acid, chlorogenic acid, neochlorogenic acid, and 4-Dicaffeoylquinic acid). Among them, hyperoside (Rt = 8.25 min), chrysoeriol 7-O-glucoside (Rt = 11.10 min), and chrysoeriol (Rt = 13.68 min) displayed deprotonated molecules at the *m/z* ratio of 463.03, 461.10, and 299.03, respectively. This is the first report of the quantification of these three flavonoids in QA that we are aware of. In addition, for the first time, the method of simultaneous quantification of 14 active components in QA using UPLC-TQ-MS/MS was reported.

According to the data in Table 4, chrysoeriol, hispidulin, eupatilin, and jaceosidin in the total extract were enriched in QA-FEA. Hyperoside, schaftoside, isoschaftoside, and six chlorogenic acid derivatives were enriched in QA-FWT after fractionation. This proves that these compounds were mostly extracted using ethyl acetate and methanol. The 14 bioactive compounds include analgesic, anti-inflammatory, and antipyretic properties that can be used to treat a variety of disorders [24,25,26]. Therefore, we can infer that the pharmacological activity of fractions depends on the content of active compounds in them.

### 2.3. Evaluation of Antioxidant Potential

The antioxidant potential of the total extract and fractions was analyzed using the DPPH colorimetric and ABTS colorimetric assays. The details are shown in Appendix A. The radical scavenging activities of the total extract, fractions, and trolox were expressed as IC50. Except for QA-FPE, all tested total extracts and fractions had a significant DPPH and ABTS scavenging potential. This may be due to the presence of phenolic compounds in QA. Hydroxyl groups in phenolic compounds react with various kinds of free radicals [27]. In the radical scavenging assay, it was understood that QA-FWT, with IC50 58.34 μg/mL (DPPH) and IC50 270.00 μg/mL (ABTS), was the most active of all the tested samples, which was lower than trolox. The antioxidant activity is closely related to the content of phenolic compounds [28]. It is known that phenolic compounds, particularly chlorogenic acids derivatives, and flavonoids are predominant in QA. Different phenolic components have different solubility in the extraction solvent (petroleum ether, ethyl acetate, and water). The antioxidant activity might be related to the majority quantities of chlorogenic acids derivatives in QA-FWT and flavonoids in QA-FEA.

### 2.4. Inhibition of the NO Release Capacity

NO release inhibition by LPS-stimulated RAW 264.7 cells was performed using five different concentrations of the total extract and fractions at 5, 10, 15, 20, and 25 μg/mL. Details are provided in the Appendix A. First, to ensure that the effects on NO release were not caused by reduced cell viability, the potential toxicity of the test materials was evaluated against RAW 264.7 cells. Samples showed cell viability of over 90%, indicating that none of the samples were harmful to the cells. Interestingly, among the samples capable of scavenging radicals, QA-FEA and QA-FWT inhibited NO significantly. Furthermore, QA-FEA showed higher activities than QA-FWT. This is because the main components in QA-FEA were flavonoids, whereas the main components in QA-FWT were chlorogenic acids. Moreover, studies have confirmed that the anti-inflammatory activities of eupatilin and jaceosidin [29] were significantly higher than chlorogenic acids. Eupatilin and jaceosidin are the main components of flavonoids enriched in QA-FEA. Inflammatory mediators are important factors to promote the occurrence of inflammation. Eupatilin and jaceosidin can effectively regulate the expression of related enzymes to inhibit the production of inflammatory mediators and prevent future inflammation. This confirms that flavonoids are more responsible for the anti-inflammatory activity of QA than chlorogenic acid derivatives.

### 2.5. Antibacterial Activities

We assessed the diameters of the inhibition zone of the total extract and three fractions against different bacteria (Appendix A). The findings are detailed in Appendix A. The diameters of the inhibition zone against *P. vulgaris* were (in ascending order) QA-FWT (17.7 mm) > QA-TE (17.3 mm) > QA-FEA (16.3 mm) > QA-FPE (13.7 mm). Similarly, the diameters of the inhibition zones against *B. subtilis* were (in ascending order) QA-FWT (20.3 mm) > QA-FEA (13.3 mm) > QA-TE (11.7 mm) > QA-FPE (10.7 mm). The diameters of the inhibition zone against S. aureus were (in ascending order) QA-FWT (22.3 mm) > QA-TE (20.7 mm) > QA-FEA (18.7 mm) > QA-FPE (14.0 mm). The diameters of the inhibition zone against *E. coli* were (in ascending order) QA-FWT (20.0 mm) > QA-TE = QA-FEA (16.7 mm) > QA-FPE (14.7 mm). The diameters of the inhibition zone against *P. aeruginosa* were (in ascending order) QA-FWT (18.7 mm) > QA-FEA (15.7 mm) > QA-TE (14.7 mm) > QA-FPE (12.7 mm). The total extract and fractions of QA inhibited two Gram-positive bacteria (*S. aureus*, *B. subtilis*) and three Gram-negative bacteria (*E. coli*, *P. aeruginosa*, *P. vulgaris*), indicating that QA has a wide antibacterial spectrum. QA-FWT had better anti-bacterial activity against different bacteria as evidenced by the diameters of the inhibition zone. This is due to the chlorogenic acid derivatives in QA that can destroy the cell wall and cell membrane structure of bacteria and certainly have an inhibitory effect on bacteria [30,31]. Beyond that, the hydroxylation at C5 and C7 of flavonoid compounds can increase the inhibition of bacterial growth [32]. The C5 and C7 of jaceosidin, eupatilin, and hispidulin riched in QA-FEA are replaced by hydroxyl groups, and the antimicrobial activity of QA-FEA is increased. This provides a theoretical basis for the application of QA as a natural antibacterial agent in food and agriculture.

## 3. Materials and Methods

### 3.1. Chemicals

3,5-di-O-caffeoylquinic acid, 4-dicaffeoylquinic acid, neochlorogenic acid, eupatilin, and 4,5-di-O-caffeoylquinic acid were purchased from Weikeqi (Chengdu, China); chrysoeriol 7-O-glucoside, chlorogenic acid, chrysoeriol, schaftoside, 3,4-di-O-caffeoylquinic acid, hispidulin, and isoschaftoside were purchased from Alfa (Chengdu, China); and hyperoside and jaceosidin were purchased from Yuanye (Shanghai, China). HPLC-grade formic acid, acetonitrile, and leucine enkephalin were purchased from Sigma-Aldrich (St. Louis, MO, USA). All other solvents (petroleum ether, ethyl acetate, methanol, ethanol) were acquired from Chron Chemicals (Chengdu, China). A Milli-Q purification system (Millipore, France) was used to create the ultra-pure water.

Dulbecco’s modified Eagle’s medium (DMEM) was purchased from Servicebio (Wuhan, China), dimethyl sulfoxide (DMSO) was purchased from Aladdin (Shanghai, China), fetal bovine serum was purchased from Newzerun (Wuhan, China), phosphate buffered saline was purchased from Hyclone (Shanghai, China), mueller hinton agar (MHA) and mueller hinton broth (MHB) were purchased from Hopebio (Qingdao, China). The DPPH Free radical Scavenging Ability assay kit and the ABTS Free radical Scavenging Ability assay kit were purchased from Jiancheng Bioengineering Institute (Nanjing, China), and the Nitric Oxide assay kit was purchased from Beyotime (Shanghai, China).

### 3.2. Plant Material

The plant samples (Figure 5) were collected from Zhulin Lake in Qichun County, Huanggang City, Hubei Province, China. The plant was collected in June 2021 and verified by Prof. Dr. Dingrong Wan, South-Central Minzu University (SCMU). Voucher specimens of Qiai plants were deposited in SCMU with the number QA2021060403. The majority of the collected plant leaves was shade dried for 7 days and then pulverized with an electric grinder to give Mugwort leaf powder.

### 3.3. Preparation of Extract and Fractions

Mugwort leaf powder (50.0 g) was extracted with 70% methanol. Extraction (1:20, *w/v*) was performed by maceration for 3 h at room temperature, heated for reflux three times in a water bath (2.5 h each time), combined with filtrate, and concentrated under vacuum to 7.6 g of the total crude extract (QA-TE). Warm water was used to dissolve 6 g of QA-TE before it was progressively partitioned with 500 mL petroleum ether (PE) and 500 mL ethyl acetate (EtOAc) to produce the PE fraction (QA-FPE, 2.0 g), EtOAc fraction (QA-FEA, 1.2 g), and water fraction (QA-FWT, 2.4 g), respectively. The extract and fractions were stored at −20 °C until use.

### 3.4. UPLC-Q-TOF/MS Analysis

Chromatographic analysis was performed on an ultra-performance liquid chromatography system equipped with a four-element pump, an online degassing machine, an automatic sampler, and a thermostatically controlled column chamber. The separation was performed on an ACQUITY UPLC HSS T3 column (100 × 2.1 mm, 1.8 µm). The mobile phase was composed of solvent A (0.1% Formic acid in H_2_O) and solvent B (0.1% Formic acid in acetonitrile: methanol, 9:1), and the elution gradient system was optimized on this basis. Elution gradient technology was used for the study, with a constant flow rate of 0.4 mL/min. The injection volume was 2 μL. The gradient proceeded as follows: 0–1.0 min, 2–5% B; 1.0–7.0 min, 5–20% B; 7.0–9.0 min, 20% B; 9.0–12.5 min, 20–28% B; 12.5–18.0 min, 28–40% B; 18.0–22.0 min, 40–98% B, 22.0–25.0 min, 98% B, 25.0–30.0 min, 98–2% B. The column and autosampler were kept at 45 and 4 °C, respectively. MS detection was carried out on Synapt-G2-SI MS system. The high collision energy ranged from 15 to 25 eV, whereas the low collision energy was fixed at 6 eV, and the ionization mode was set as ESI^+^ and ESI^−^. The mass ranged from 50 to 1200 Da. The cone voltage was 40 V, the capillary voltage was 3.00 kV in the negative mode and 2.59 kV in the positive mode. The desolvation temperature was fixed at 500 °C, while the ion source temperature remained at 150 °C. Desolvation gas (N2) flowed at 800 L/h while cone gas (N2) flowed at 50 L/h.

### 3.5. Construction of UNIFI Theoretical Library on Chemical Constituents of QA

SciFinder, PubMed, PubChem, and Reaxys are a few of the internet databases that were used to compile a list of the compounds mentioned in the literature on QA. Search terms “*Artemisia argyi*” were employed to search published literature up to April 2022. The process of identifying chemical structures in complex natural products can be streamlined by combining UPLC-Q-TOF/MS data with the UNIFI information management platform and its embedded Traditional Medicine Library. Finally, the structure of 208 compounds reported from *A. argyi* species was collected and saved in a .sdf file as a theoretical library. The MS data of the QA-TE was imported into the UNIFI platform for rapid matching screening with the theoretical library data of *A. argyi* compounds.

### 3.6. UPLC-TQ-MS/MS Quantitative Analysis of Main Components

#### 3.6.1. Preparation of Standard Solution and Sample Solution

Flavonoids and chlorogenic acids are important components in QA, which are closely related to the pharmacological action of QA. Therefore, it is significant to quantify the main flavonoids and chlorogenic acids in QA.

A mixed standard stock solution containing hyperoside, chrysoeriol 7-O-glucoside, chlorogenic acid, chrysoeriol, schaftoside, 3,4-di-O-caffeoylquinic acid, 3,5-di-O-caffeoylquinic acid, hispidulin, jaceosidin, 4-dicaffeoylquinic acid, neochlorogenic acid, eupatilin, isoschaftoside, and 4,5-di-O-caffeoylquinic acid was prepared in methanol:water (1:1, *v/v*). To prepare working standard solutions for plotting the calibration curve, mixed standards were diluted with methanol within the ranges from 3.2 to 1000 ng/mL.

A total of 2–3 mg samples were taken, QA-TE was dissolved in methanol:water (1:1), and QA-FEA and QA-FWT were dissolved in methanol. The sample solution was centrifuged with a centrifuge (Eppendorf 5810R) at 10,000 r/min, and the supernatant was used for the test. The QA-TE and QA-FWT were diluted to 50 μg/mL and the QA-FEA to 10 μg/mL.

#### 3.6.2. Instrumentation and Analytical Conditions

Chromatographic analysis was the same as 2.4. The Xevo TQ-S MS/MS system was used to perform the mass spectrometry detection. The ionization mode for was set to ESI^+^ and ESI^−^ mode for the determination of the main chemical constituents of QA by the UNIFI theoretical library. The quantitative data acquisition mode was set to multiple reaction monitoring (MRM), the ionization mode was set to ESI^−^, and the other analysis conditions of mass spectrometry were consistent with 2.4. Each analyte’s collision energy and particular fragmentor voltage were tuned in order to produce the strongest quantitative change. Appendix A in the supplementary document includes the optimum values for these critical parameters for the fourteen target compounds.

### 3.7. Evaluation of Antioxidant Activity

#### 3.7.1. DPPH Assay

The scavenging activities of the total extract and three fractions were evaluated using a 2.2-dy-phenyl-1-picrylhydrazyl (DPPH) Free Radical Scavenging Ability Assay kit with slight modifications [33]. DPPH (600 μL) was admixed with 400 μL of fractions and standard (4.0–426.0 μg/mL), respectively. After being vortexed, the reaction mixture was left at room temperature in the dark for 30 min. After incubation, absorbance was assessed at 517 nm using a spectrophotometer. Methanol was employed as a blank, and trolox served as the positive control (standard). Each blank, samples, and standards’ absorbance were measured in triplicate. The ability to scavenge the DPPH radical was measured by the following equation:%DPPH radical scavenging = (1 − (A_i_ − A_j_) ÷A_0_) × 100%

A_i_: absorbance of DPPH radical + fraction/standard; A_j_: absorbance of fraction/standard + methanol;

A_0_: absorbance of DPPH radical + methanol.

By graphing the sample concentration vs. the scavenging capacity using a logarithm function, the IC50 (Half-maximal Inhibitory Concentration) value was determined.

#### 3.7.2. ABTS Assay

The scavenging activity of the total extract and three fractions was evaluated using a 2,2′-Azino-bis-3-ethylbenzothiazoline-6-sulphonic acid (ABTS) Free Radical Scavenging Ability Assay kit. The detection buffer, ABTS solution, and hydrogen peroxide solution (76:5:4) were mixed to prepare the ABTS working solution. Trolox was used as a positive control (standard). ABTS (170 μL), and peroxidase solution (20 μL) were admixed with 10 μL of fractions and standard (51.8–837.0 μg/mL), respectively. The reaction mixture was vortexed and left at room temperature in the dark for 6 min. After incubation, absorbance was measured by an enzyme standard instrument at 405 nm. The ability to scavenge the ABTS radical was measured by the following equation:%DPPH radical scavenging = (A_0_ − A_i_) ÷ A_0_ × 100%

A_i_: absorbance of ABTS radical + peroxidase solution+ fraction/standard;

A_0_: absorbance of ABTS radical + peroxidase solution+ H_2_O.

### 3.8. Determination of Anti-Inflammatory Activity by Inhibition of NO

The inhibiting effect on nitric oxide (NO) production in LPS-stimulated RAW 264.7 (Wuhan, China) macrophage cells served as a metric for the anti-inflammatory action. The cells were cultured in Dulbecco’s modified Eagle’s medium (DMEM) supplemented with 10% (*v/v*) fetal bovine serum and 0.5% penicillin/streptomycin. The cells were cultivated in a humidified incubator at 37 °C with 5% CO_2_ and 95% air. Measurements were made of the samples’ ability to inhibit NO generation. In 96-well culture plates filled with 100 L of DMEM media, RAW 264.7 cells (6 × 10^4^) were planted. After 2 h of cell adhesion, the cells were starved for 12 h. LPS (1 μg/mL) and different concentrations of sample solution (25, 20, 15, 10, 5 μg/mL) were added simultaneously. The cells were incubated at 37 °C with 5% CO_2_ for 24 h. After 24 h of incubation, 50 μL of the supernatant was collected for nitrite assay with a NO assay kit by using the Griess reaction [34]. The remaining medium was taken out, and the CCK-8 technique was used to assess the cell viability. The absorbance was measured at 450 nm.

### 3.9. Disc Diffusion Assay

The agar plates’ preparation was performed for the disc diffusion technique to examine the antibacterial activity of the extract and fractions. Two Gram-positive bacteria (*Staphylococcus aureus*, *Bacillus subtilis*) and three Gram-negative bacteria (*Escherichia coli*, *Pseudomonas aeruginosa*, *P. vulgaris*) were chosen for antibacterial activities of the total extract and fractions. Each strain was cultivated for 24 h, and the bacterial culture was diluted to a concentration of about 10^6^ CFU/mL. A total of 0.2 mL of the diluted solution was then evenly dispersed over the agar plates. Samples were diluted with methanol at 50 mg/mL. Then, 0.2 mL of the sample solution was injected into a 6 mm diameter hole placed in the agar plates. The plates were cultured at 37 °C for 16 h. To assess the antibacterial activity of the strains, the widths of their inhibition zones were evaluated. Methanol (ME) was used as a negative control, and 5 μg of ciprofloxacin hydrochloride (CH) was used as a positive control.

## 4. Conclusions

In conclusion, this study established a rapid identification method for compounds in QA by combining UPLC-Q-TOF/MS with the UNIFI information management platform. Meanwhile, the study provided an effective method for the quantitative analysis of 14 compounds in QA by UPLC-TQ-MS/MS. This method could quantify 14 compounds simultaneously and be verified by LODs, LOQs, precision, repeatability, stability, and recovery range. The QA-FEA obtained from the QA-TE significantly reduced the NO release by LPS-stimulated RAW 264.7 cells. Meanwhile, QA-FWT has the highest DPPH and ABTS free radical scavenging ability and antibacterial ability. This is because QA-FEA has the highest flavonoid content and QA-FWT has the highest phenolic acid content. The results showed that *Artemisia argyi* Lévl. et Van., as dietary and traditional Chinese medicine, was an excellent source of natural antioxidants, anti-inflammatory drugs, and antibacterial agents. The results provided the theoretical basis for the use of QA in the food and pharmaceutical industries. The plant material selected for this study was from one production area, so there are some limitations. Factors such as geographical location, variety, and climate can have significant effects on the chemical composition of *Artemisia argyi* Lévl. et Van. In the future, we will work to improve the information on the chemical composition of *Artemisia argyi* in terms of different cultivars and origins to provide more comprehensive and reliable information for the research and application of *Artemisia argyi*.

## Figures and Tables

**Figure 1 molecules-28-02022-f001:**
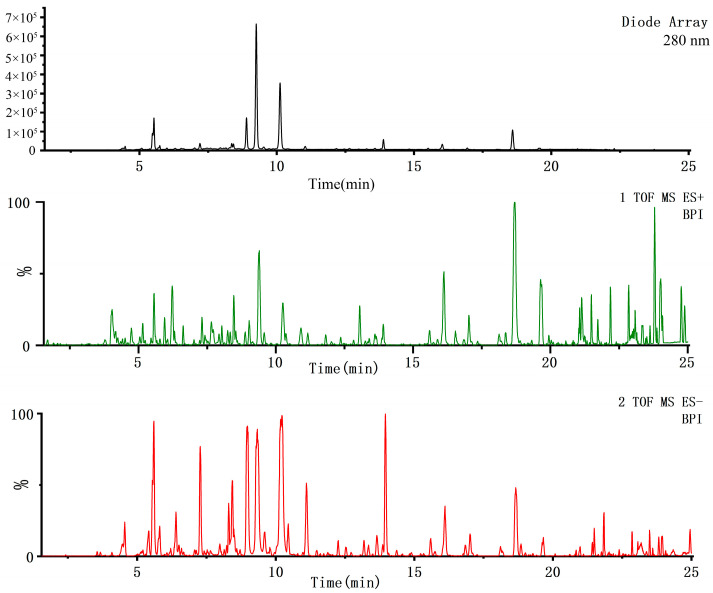
The representative chromatogram and base peak ion chromatograms (BPI) in positive and negative ions of QA.

**Figure 2 molecules-28-02022-f002:**
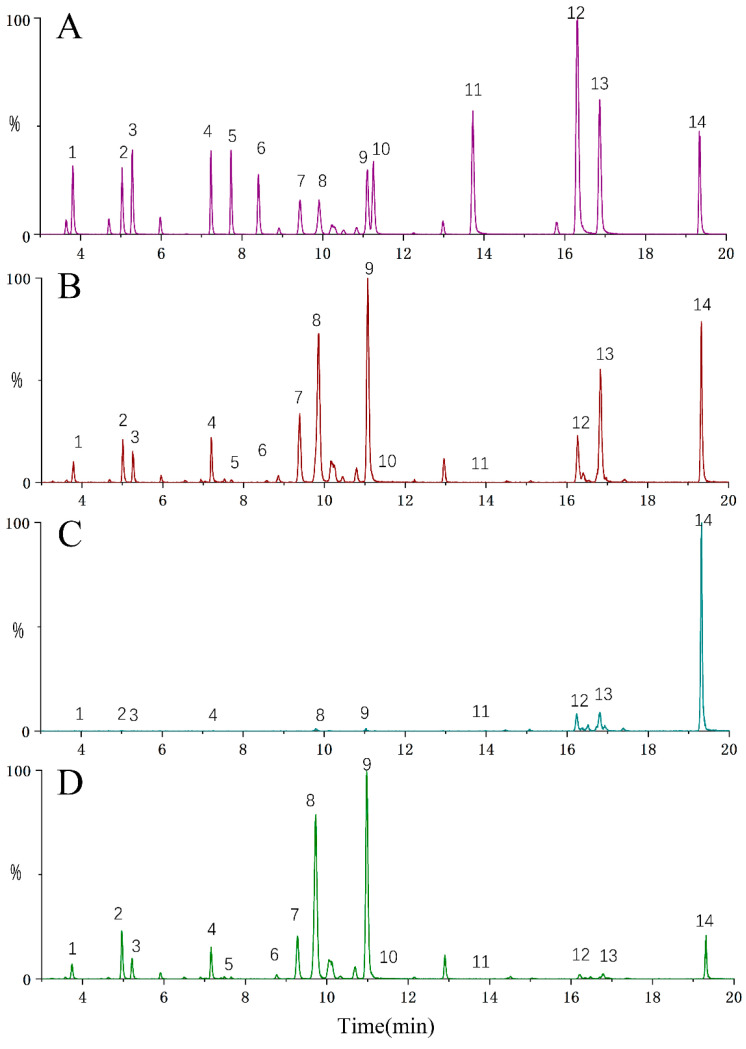
Representative UPLC-TQ-MS/MS total ion chromatogram of standards (**A**), QA-TE (**B**), QA-FEA (**C**), and QA-FWT (**D**).

**Figure 3 molecules-28-02022-f003:**
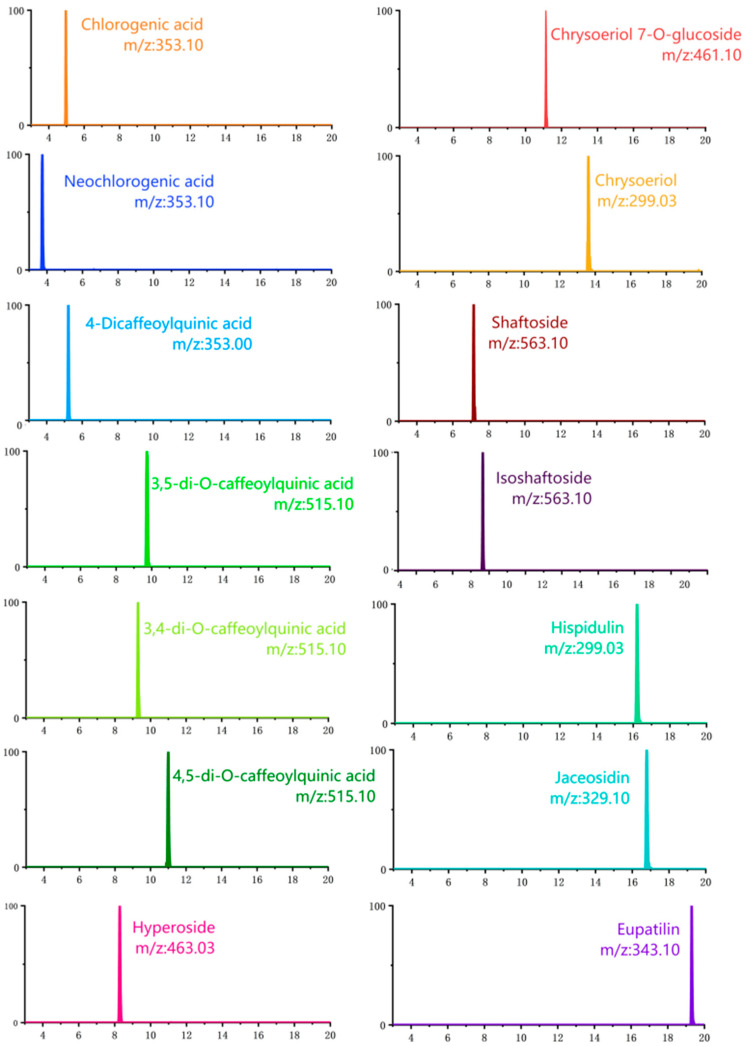
The ion chromatograms of 14 standards under the optimal UPLC-TQ-MS/MS conditions.

**Figure 4 molecules-28-02022-f004:**
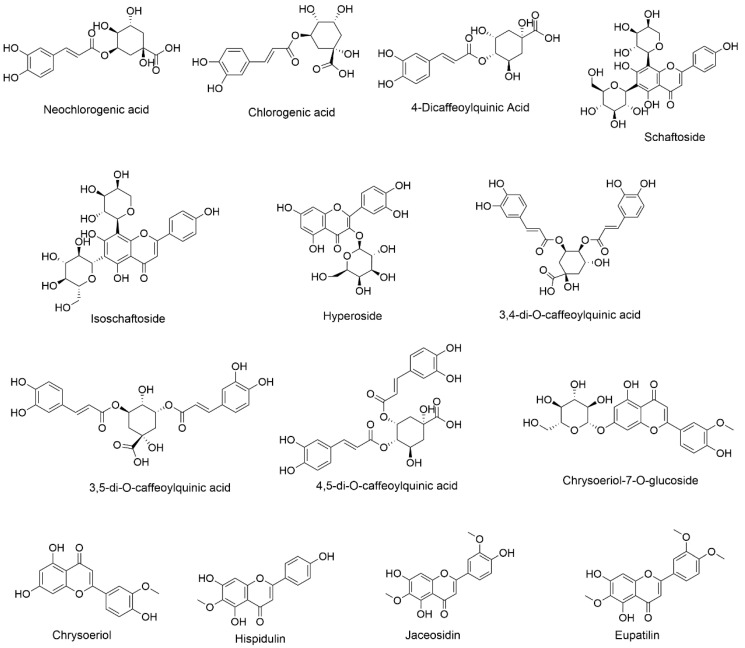
The structures of quantified compounds in Qiai.

**Figure 5 molecules-28-02022-f005:**
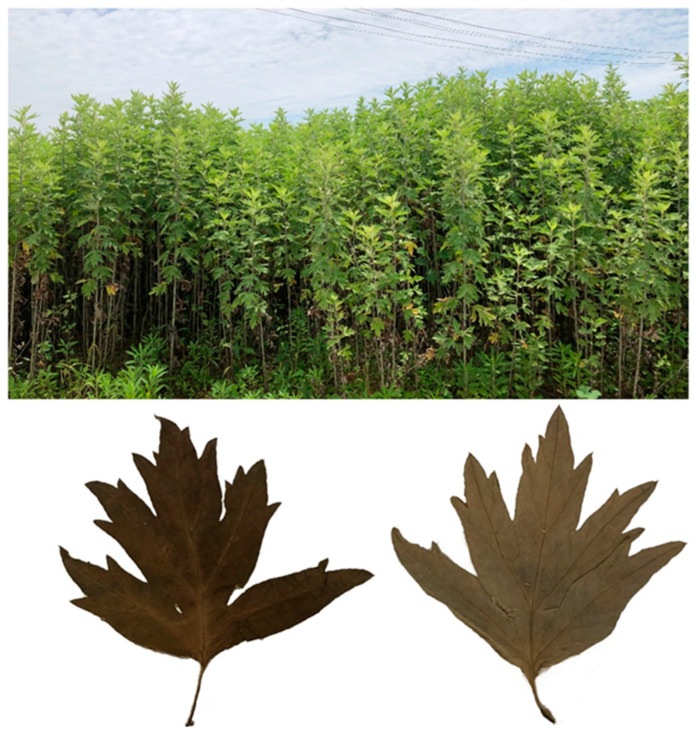
A picture of Qiai.

**Table 1 molecules-28-02022-t001:** Tentatively identified major metabolites from BPI chromatograms of QA (in positive mode).

No.	Component Name	Observed RT (min)	Formula	Observed Neutral Mass (Da)	Observed *m/z*	Mass Error (mDa)	Adducts
1	14-deoxyactucin	1.65	C_15_H_16_O_4_	260.1047	278.1386	−0.1	+NH_4_
2	Artemisargins B	3.99	C_18_H_24_O_7_	376.1522	394.1860	0	+NH_4_
3	Neochlorogenic acid	4.47	C_16_H_18_O_9_	354.0938	353.0910	0.2	+H
4	Arteglasin A	4.57	C_17_H_20_O_5_	304.1307	322.1645	−0.4	+NH_4_
5	7-hydroxy-2H-chromen-2-one	5.53	C_9_H_6_O_3_	162.0323	163.0396	0.6	+H
6	Moxartenolide	6.98	C_20_H_22_O_5_	342.1462	360.18	−0.5	+NH_4_
7	Schaftoside	7.27	C_26_H_28_O_14_	564.1481	565.1554	0.2	+H
8	5alpha-hydroxydehydroleucodin	7.28	C_15_H_16_O_4_	260.1049	261.1122	0	+H
9	Argyin D	7.28	C_15_H_18_O_5_	278.1153	301.1045	−0.1	+Na
10	Austroy unnane D	7.63	C_15_H_18_O_5_	278.1155	301.1047	0.1	+Na
11	7-hydroxy-6-methoxy-2H-1-benzopyran-2-one	8.01	C_10_H_8_O_4_	192.0423	193.0496	0	+H
12	10-epi-artecanin	8.09	C_15_H_18_O_5_	278.1151	301.1044	−0.3	+Na
13	Eriodictyol	8.37	C_15_H_12_O_6_	288.0633	289.0705	−0.1	+H
14	Quercetol	8.43	C_15_H_10_O_7_	302.0432	303.0505	0.6	+H
15	Hyperoside	8.43	C_21_H_20_O_12_	464.0949	465.1021	−0.6	+H
16	Luteolin	8.50	C_15_H_10_O_6_	286.0479	287.0552	0.2	+H
17	13-acetoxy-8alpha-hydroxy-7,11-dehydro-11,13-dihydroanhydrovertorin	8.56	C_17_H_22_O_6_	322.1413	323.1486	−0.4	+H
18	3,4-di-O-caffeoylquinic acid	8.99	C_25_H_24_O_12_	516.1263	517.1336	−0.4	+H
19	Demethoxy aschantin	9.09	C_19_H_18_O_4_	310.1199	328.1537	−0.6	+NH_4_
20	3,5-di-O-caffeoylquinic acid	9.35	C_25_H_24_O_12_	516.1262	517.1335	−0.6	+H
21	Tuberiferine	10.87	C_15_H_18_O_3_	246.1253	247.1325	−0.3	+H
22	Trichocadinin C	10.88	C_15_H_16_O_4_	260.1043	261.1116	−0.6	+H
23	Argyin C	13.57	C_19_H_24_O_7_	364.1524	365.1597	0.2	+H
24	5,6,2′,4′-tetrahydroxy-7,5′-dimethoxyflavone	13.83	C_17_H_14_O_8_	346.0689	347.0761	0	+H
25	Eupafolin	15.56	C_16_H_12_O_7_	300.0637	301.071	0.4	+H
26	Hispidulin	15.56	C_10_H_8_O_3_	300.0637	301.071	0.4	+H
27	Trichocadinin B	15.85	C_15_H_16_O_3_	244.1101	245.1174	0.2	+H
28	Jaceosidin	16.08	C_17_H_14_O_7_	330.0749	331.0822	0.9	+H
29	Artemisian D	16.10	C_30_H_36_O_8_	524.2414	542.2752	0.3	+NH_4_
30	Artemisiane B	16.20	C_30_H_34_O_9_	538.2197	561.2089	−0.6	+Na
31	Artemisian A	16.25	C_30_H_36_O_8_	524.2408	542.2746	−0.2	+NH_4_
32	Apicin	17.00	C_18_H_16_O_8_	360.0847	361.0920	0.2	+H
33	Jaceidin	17.00	C_18_H_16_O_8_	360.0847	361.0920	0.2	+H
34	5,7-dihydroxy-3′,4′-dimethoxy flavone	18.09	C_17_H_14_O_6_	314.0791	315.0864	0.1	+H
35	5,6-dihydroxy-3′,4′,7-trimethoxyflavone	18.66	C_18_H_16_O_7_	344.0901	345.0973	0.5	+H
36	Eupatilin	18.66	C_18_H_16_O_7_	344.0901	345.0973	0.5	+H
37	Artanomaloide	18.86	C_30_H_34_O_7_	506.2302	507.2375	−0.2	+H
38	Artemisianin A	18.86	C_30_H_36_O_8_	524.2409	542.2747	−0.1	+NH_4_
39	Chrysoplenitin	19.62	C_19_H_18_O_8_	374.1007	375.1079	0.5	+H
40	Ladanein	19.99	C_17_H_14_O_6_	314.0789	315.0862	−0.1	+H
41	8-acetylarteminolide	20.03	C_32_H_36_O_8_	548.2403	549.2476	−0.7	+H
42	Artemetin	20.79	C_20_H_20_O_8_	388.1154	389.1227	−0.4	+H
43	Koninginin T	21.46	C_17_H_26_O_3_	276.2086	277.2159	−0.3	+H
44	Artanomaloide C	21.46	C_35_H_40_O_8_	588.2722	589.2795	−0.1	+H
45	9-oxo-(10E,12E)-octadeca-10,12-dienoic acid	22.00	C_18_H_30_O_3_	294.2189	317.2081	−0.6	+Na
46	Artemisolide	22.15	C_25_H_32_O_4_	396.2303	397.2376	0.2	+H
47	Artemargyinolide A	22.36	C_40_H_50_O_7_	642.3552	660.389	−0.5	+NH_4_

**Table 2 molecules-28-02022-t002:** Tentatively identified major metabolites from BPI chromatograms of QA (in negative mode).

No.	Component Name	Observed RT (min)	Formula	Observed Neutral Mass (Da)	Observed *m/z*	Mass Error (mDa)	Adducts
1	Cirsilineol	4.51	C_17_H_14_O_7_	330.0715	375.0697	−2.5	+HCOO
2	4-Dicaffeoylquinic Acid	5.76	C_16_H_18_O_9_	354.0957	353.0885	0.6	-H
3	Acrifolide	6.54	C_15_H_16_O_6_	292.0947	337.0929	0	+HCOO
4	1β,2β-epoxy-3β,4α,10α-trihydroxyguaian-6α,12-olide	6.63	C_15_H_20_O_6_	296.1262	295.1189	0.2	-H
5	Isotanciloide	6.63	C_15_H_20_O_6_	296.1262	295.1189	0.2	-H
6	Schaftoside	7.24	C_26_H_28_O_14_	564.1494	563.1421	1.5	-H
7	Isoschaftoside	7.48	C_26_H_28_O_14_	564.1486	563.1413	0.7	-H
8	Argyin D	7.61	C_15_H_18_O_5_	278.116	277.1088	0.6	-H
9	10-epi-artecanin	7.95	C_15_H_18_O_5_	278.116	277.1087	0.6	-H
10	Artemetin	8.10	C_20_H_20_O_8_	388.115	433.1132	−0.8	+HCOO
11	Hyperoside	8.39	C_21_H_20_O_12_	464.0971	463.0899	1.7	-H
12	3alpha,4alpha,10beta-trihydroxy-8alpha-acetoxyguai-1,11(13)-dien-6alpha,12-olide	8.59	C_17_H_22_O_7_	338.1368	337.1295	0.2	-H
13	3,4-di-O-caffeoylquinic acid	8.93	C_25_H_24_O_12_	516.1278	515.1206	1.1	-H
14	Chlorogenic acid	9.29	C_16_H_18_O_9_	354.0953	353.088	0.2	-H
15	4,5-di-O-caffeoylquinic acid	9.29	C_25_H_24_O_12_	516.1275	515.1203	0.8	-H
16	4-Hydroxyacetophenone	10.16	C_8_H_8_O_2_	136.0529	135.0456	0.5	-H
17	3,4-O-dicaffeoylquinic acid	10.16	C_25_H_24_O_12_	516.1285	515.1212	1.7	-H
18	Chrysoeriol 7-O-glucoside	11.08	C_22_H_22_O_11_	462.3601	461.3528	0.8	-H
19	Eriodictyol	12.68	C_15_H_12_O_6_	288.0638	287.0565	0.4	-H
20	Eupatilin 7-O-beta-D-glucopyranoside	13.08	C_24_H_26_O_12_	506.1433	551.1415	0.8	+HCOO
21	Luteolin	13.31	C_15_H_10_O_6_	286.0484	285.0411	0.7	-H
22	Apigenin	13.61	C_15_H_10_O_5_	270.0534	315.0516	0.6	+HCOO
23	Chrysoeriol	13.67	C_16_H_12_O_6_	300.2678	299.2909	1.8	-H
24	5,6,2′,4′-tetrahydroxy-7,5′-dimethoxyflavone	13.82	C_17_H_14_O_8_	346.0692	345.0619	0.3	-H
25	Naringenin	14.86	C_15_H_12_O_5_	272.0686	271.0613	0.1	-H
26	Hispidulin	15.55	C_16_H_12_O_6_	300.064	299.0567	0.6	-H
27	Eupafolin	15.7	C_16_H_12_O_7_	300.0639	299.0566	0.5	-H
28	Jaceidin	16.01	C_18_H_16_O_8_	360.0845	359.0772	0	-H
29	Jaceosidin	16.07	C_17_H_14_O_7_	330.0745	329.0672	0.5	-H
30	Artemisian D	16.08	C_30_H_36_O_8_	524.2411	569.2393	0.1	+HCOO
31	Artemisian A	16.23	C_30_H_36_O_8_	524.2403	523.233	−0.7	-H
32	5,7-dihydroxy-3′,4′-dimethoxy flavone	16.98	C_17_H_14_O_6_	314.0796	359.0778	0.6	+HCOO
33	Apicin	16.98	C_18_H_16_O_8_	360.0851	359.0778	0.6	-H
34	Ladanein	18.07	C_17_H_14_O_6_	314.0796	313.0723	0.6	-H
35	5,6-dihydroxy-3′,4′,7-trimethoxyflavone	18.63	C_18_H_16_O_7_	344.0905	343.0833	0.9	-H
36	Eupatilin	18.63	C_18_H_16_O_7_	344.0905	343.0833	0.9	-H
37	Artemisian C	18.81	C_30_H_36_O_8_	524.2418	523.2345	0.8	-H
38	Artemisianin D	18.97	C_30_H_36_O_8_	524.2416	523.2343	0.5	-H
39	Chrysoplenitin	19.61	C_19_H_18_O_8_	374.1009	373.0936	0.7	-H
40	Argyinolide O	20.58	C_30_H_34_O_6_	490.2362	535.2344	0.7	+HCOO
41	13-oxo-9Z,11E-octadecadienoic acid	21.45	C_18_H_30_O_3_	294.2204	293.2131	0.9	-H
42	Artanomaloide A	21.83	C_35_H_42_O_8_	590.2895	635.2877	1.5	+HCOO
43	Artemilinin A	22.82	C_30_H_40_O_7_	528.3065	527.2992	−2.2	-H

**Table 3 molecules-28-02022-t003:** The regression equation, linear range, LOD, LOQ, intra-day and inter-day precision, and recovery of the developed UPLC-TQ-MS/MS method.

No.	Standards	Regression Equation	Linear Range (ng/mL)	R^2^	LOD (ng/mL)	LOQ (μg/mL)	Intra-Day RSD (%)	Inter-Day RSD (%)	Recovery Range (%)
1	Neochlorogenic acid	y = 81.359x − 414.46	1.60–2000.00	0.9999	0.50	1.52	1.34	0.87	99.74 ± 2.03
2	Chlorogenic acid	y = 112.05x − 54.224	1.60–2000.00	0.9999	0.49	1.48	1.02	1.38	98.61 ± 1.02
3	4-Dicaffeoylquinic Acid	y = 86.488x − 151.05	3.20–2000.00	0.9999	1.04	3.15	1.28	0.99	99.26 ± 1.47
4	Schaftoside	y = 63.919x − 81.799	3.20–2000.00	0.9997	1.05	3.18	2.03	1.98	101.74 ± 1.77
5	Isoschaftoside	y = 66.869x − 204.17	1.60–2000.00	0.9999	0.50	1.52	1.57	1.35	98.81 ± 1.56
6	Hyperoside	y = 151.14x − 129.05	1.60–2000.00	0.9999	0.51	1.55	1.18	1.36	98.26 ± 1.63
7	3,4-di-O-caffeoylquinic acid	y = 70.087x − 1247.2	1.60–2000.00	0.9999	0.48	1.45	0.86	1.26	97.56 ± 1.04
8	3,5-di-O-caffeoylquinic acid	y = 82.844x − 1638.1	8.00–2000.00	0.9984	2.63	7.97	0.79	1.01	98.62 ± 0.98
9	4,5-di-O-caffeoylquinic acid	y = 152.35x − 2803.6	8.00–2000.00	0.9988	2.61	7.91	1.33	1.21	100.85 ± 0.97
10	Chrysoeriol 7-O-glucoside	y = 325.22x + 4607.7	3.20–2000.00	0.9978	1.05	3.18	1.55	1.16	99.57 ± 1.62
11	Chrysoeriol	y = 311.26x + 2755.2	1.60–2000.00	0.9996	0.49	1.48	1.96	1.73	99.08 ± 2.06
12	Hispidulin	y = 636.33x + 22250	1.60–2000.00	0.9967	0.49	1.48	1.94	2.16	98.63 ± 1.29
13	Jaceosidin	y = 406.28x + 3851.4	16.00–2000.00	0.9996	5.32	15.89	2.31	2.07	97.93 ± 1.59
14	Eupatilin	y = 191.01x + 1368	8.00–2000.00	0.9997	2.63	7.97	0.96	1.41	99.37 ± 1.66

**Table 4 molecules-28-02022-t004:** Quantitative analytical results for the 14 compounds in extracts and fractions of QA (n = 3).

Name	Content (mg/g)		
	QA-TE	QA-FEA	QA-FWT
Neochlorogenic acid	3.59 ± 0.07 **	0.07 ± 0.01 *	3.52 ± 0.06 *
Chlorogenic acid	16.67 ± 0.21 **	0.40 ± 0.01 *	16.67 ± 0.19 **
4-Dicaffeoylquinic Acid	2.79 ± 0.09 **	0.06 ± 0.01 *	3.74 ± 0.06 *
Schaftoside	2.68 ± 0.04 *	0.16 ± 0.01 *	5.98 ± 0.05 **
Isoschaftoside	0.10 ± 0.01 *	—	0.27 ± 0.01 *
Hyperoside	0.34 ± 0.03 *	0.13 ± 0.01 *	1.32 ± 0.05 *
3,4-di-O-caffeoylquinic acid	19.30 ± 0.23 **	—	69.16 ± 0.71 **
3,5-di-O-caffeoylquinic acid	93.63 ± 1.15 **	8.21 ± 0.33 **	101.40 ± 1.83 **
4,5-di-O-caffeoylquinic acid	48.27 ± 0.46 **	3.35 ± 0.11 *	73.55 ± 0.68 **
Chrysoeriol 7-O-glucoside	0.15 ± 0.02 *	—	0.08 ± 0.01 *
Chrysoeriol	0.01 ± 0.01 *	0.04 ± 0.01 *	0.01 ± 0.01 *
Hispidulin	0.57 ± 0.01 *	1.93 ± 0.02 **	0.35 ± 0.01 *
Jaceosidin	2.41 ± 0.04 **	15.61 ± 0.45 **	2.79 ± 0.02 **
Eupatilin	7.99 ± 0.29 **	48.54 ± 0.77 **	6.90 ± 0.08 **

* 0.01 < *p* < 0.05, ** *p* < 0.001.

## Data Availability

Data is contained within the article or Appendix A.

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
