# Peer review of "UPLC-MS Analysis, Quantification of Compounds, and Comparison of Bioactivity of Methanol Extract and Its Fractions from Qiai (Artemisia argyi Lévl. et Van.)"

_molecules, 2023, doi:10.3390/molecules28052022_

Round 1
Reviewer 1 Report
Comments
1- Should write two or three sentences regarding the plant and its previous investigations
2- Check the a previous article published (Rapid Identification of Chemical Constituents in Artemisia argyi Le´vi. et Vant by UPLC-Q-Exactive-MS/MS) and cite it as it has previous identification of some compounds from the same species.
3- Figure 1 should be in the material and method section
4- The antibacterial studies used Disc diffusion assay (please provide the photos of the discs.
Author Response
请参阅附件。

Reviewer 2 Report
The manuscript “UPLC-MS Analysis, Quantification of Compounds, and Comparison of Bioactivity of Methanol Extract and its Fractions from Qiai (Artemisia argyi Lévl. et Van.)” was focused on qualitative and quantitative analysis of phytochemicals, as well as estimation of antioxidant, anti-inflammatory and antibacterial activities of the methanolic extract and its fractions of Artemisia argyi. Through using HRMS combined with databases, more than 60 phytochemicals were tentatively identified, of which 14 were quantified. Ethyl acetate fraction exhibited the most potent anti-inflammatory activity, while the aqueous fraction showed the highest antioxidant and antibacterial activities. Through reviewing the manuscript, I recommend reconsidering the following minor points:
- Section 3.3: please clarify how much petroleum ether and ethyl acetate were used in the fractionation step.
- Table 4: statistically significant differences in phytochemical concentrations among the extracts should be noted.
- Did the authors think peak #14 (Figure 3) should be quantified as a compound due to the two coeluting #35 and 36 (Table 1,2)?
Reviewer 3 Report
The abstract should contain 1-2 promising compounds with the best antioxidant/antibacterial activity. This will help readers to understand the manuscript better.
The conclusion should include the limitations of the study.
There are some minor typographical errors, and these should be corrected.
Round 2
Reviewer 1 Report
the authors corrected all the required comments